# Predicting the Content of 20 Minerals in Beef by Different Portable Near-Infrared (NIR) Spectrometers

**DOI:** 10.3390/foods9101389

**Published:** 2020-10-01

**Authors:** Nageshvar Patel, Hugo Toledo-Alvarado, Alessio Cecchinato, Giovanni Bittante

**Affiliations:** Department of Agronomy, Food, Natural resources, Animals and Environment (DAFNAE), University of Padova, 35020 Legnaro, Italy; h.toledo.a@comunidad.unam.mx (H.T.-A.); alessio.cecchinato@unipd.it (A.C.); bittante@unipd.it (G.B.)

**Keywords:** micro-nutrients, trace and environmental elements, portable NIR instruments, near infrared absorbance, chemometrics

## Abstract

The aim of this study was to test the predictability of a detailed mineral profile of beef using different portable near-infrared spectrometers (NIRS). These devices are rapid, chemical waste-free, cheap, nondestructive tools that can be used directly on the meat surface in the work environment without the need to take samples. We compared a transportable Visible-NIRS (weight 5.6 kg; wavelength 350–1830 nm), a portable NIRS (2.0 kg; 950–1650 nm), and a hand-held Micro-NIRS (0.06 kg; 905–1649 nm) to predict the contents of 20 minerals (measured by ICP-OES) in 178 beef samples (*Longissimus thoracis* muscle) using different mathematical pretreatments of the spectra and partial least square regressions. The externally validated results show that Fe, P, Mg, S, Na, and Pb have some potential for prediction with all instruments (*R*^2^_VAL_: 0.40–0.83). Overall, the prediction performances of the three instruments were similar, although the smallest (Micro-NIRS) exhibited certain advantages.

## 1. Introduction

Minerals are important nutrients in meat, an important human food. Furthermore, the contents of some minerals are related to meat quality [1], while others are indicators of environmental pollution [2]. Therefore, information of the mineral profile of beef, which is affected by several factors, such as animal breed and farm management practices, including feeding regime, is important for the production of commercial beef of a desirable quality, and to understand the environmental impact of beef production chains.

The mineral content of meat is usually determined by various chemical analytical methods [3], such as atomic absorption spectrometry [4], inductively coupled plasma–optical emission spectrometry (ICP-OES), inductively coupled plasma–mass spectrometry (ICP-MS) [5], and neutron activation analysis (NAA) [6]. A few other methods, e.g., X-ray emission spectrometry, molecular light absorption spectrometry, and molecular fluorometry, are also used to analyze minerals in food and feed materials [7]. These techniques are accurate and precise, but they are time consuming, require sample preparation, use costly reagents and apparatuses, and produce toxic waste. Abattoirs and meat processing companies are therefore seeking easy, rapid, non-destructive, albeit less accurate, methods that could be used directly on meat carcasses and cuts.

One such technique is near-infrared spectroscopy (NIRS), which is well-known for being a rapid, non-invasive, non-destructive means of measuring multiple beef quality traits, and for being ecofriendly and inexpensive [8]. It is used for predicting beef chemical composition [9,10], and for determining fatty acid profiles [11] and physical traits, such as color, tenderness, and drip and cooking water losses [12,13]. NIRS is a method that is sensitive to hetero atomic covalent bonds; therefore, it is not sensitive to metals. Minerals in fact are not or weakly associated with any specific bands in the infrared part of the electromagnetic radiation absorption spectrum, but the concentrations of some minerals can be predicted using proper chemometric treatment of the whole spectrum if they are bounded to some organic molecule [3]. Often these minerals are those more interesting for animal physiology, as well as for human nutrition, because they have important roles in many metabolic pathways [2].

NIRS is widely diffused in the industry for its rapidity, simplicity, and inexpensiveness to analyze the chemical content of many organic molecules, predict some physical properties, or monitor the food processing chain. Thus, even though it has many limitations for analyzing minerals in food, it could be interesting to add the prediction of the content of some minerals to other analyses already monitored by the industry using NIRS. Few studies have been made on NIRS-based predictions of meat mineral contents; these have dealt with pork [14], mutton [15], and chicken [16], but none that the authors are aware of has dealt with beef. Moreover, the majority of the studies on NIRS predictions of minerals in food and feed materials were carried out with benchtop instruments, which require samples to be taken and then transferred to the laboratory for analysis, and often require the samples to be preprocessed before the spectra are acquired. However, these problems could be overcome by the use of robust, portable NIR instruments to collect spectra directly on the meat surface at the abattoir or processing plant without the need to take meat samples. Rapid developments in the field of optics, and advances in the miniaturization of infrared detector components mean that cheaper, transportable/portable/hand-held NIR spectrometers for instant analysis are now available [13,17,18]. However, the reduction in size presents a challenge to spectral performance in terms of wavelength range, resolution, and signal-to-noise ratio. We postulate that portable NIRS can be applied directly to the muscle surface, with no sample preparation, to predict the contents of some minerals in beef, but, given their wide variation in size, portability, and technical features, the instruments currently available need to be tested.

The objectives of this study were: (a) to test the ability of portable infrared spectrometers to predict 20 macro-, essential micro-, and environmental (trace) micro-minerals in beef; and (b) to compare the prediction ability of three spectrometers of different sizes, spectral ranges and intensities, technical features, and costs through external validation.

## 2. Materials and Methods

### 2.1. Experimental Design, Animals and Samples

Details of the experimental design and the selection of farms and animals for the trial were given in a previous study within this project [19]. The experimental design of this work is summarized in Table 1.

As a case-study, we analyzed the beef production system of the PGI certified “Vitellone Bianco dell’Appennino Centrale” (Central Apennine White Young Bull). Ninety-one young bulls and heifers of the Chianina and Romagnola breeds, reared, fed, and slaughtered according to the PGI norms (European Union regulation 134/1998), were randomly selected from 15 farms and sampled. The farms were located in the historical areas of origin of the Chianina (mainly Tuscany/Umbria) and the Romagnola (Emilia-Romagna) breeds, and were selected by the “Consorzio Produttori Carne Bovina Pregiata delle Razze Italiane” (CCBI, Consortium of Producers of High-Quality Beef from Italian Breeds), which is responsible for controlling and monitoring the PGI certification.

Briefly, 182 rib cuts were taken from 91 carcasses (one from the right and one from the left side) at the level of the 5th rib, in accordance with European Commission regulations (EC 1249/2008, EEC1208/81), the day after slaughter. The cuts were cooled, vacuum packed, and labeled, and then taken to the Meat Laboratory of the Department of Agronomy, Food, Natural resources, Animals and Environment (DAFNAE) of the University of Padova, Italy, for analysis.

### 2.2. Portable Near-Infrared Instruments

We selected three portable near-infrared spectrometers of different sizes and with different technical characteristics, levels of portability, and spectrum ranges. Details of the technical and physical features of these instruments are presented in Table 2. The transportable visible near-infrared spectrometer (Vis-NIRS), the most expensive of the three, uses both visible and infrared wavelengths (350–1830 nm) in 1-nm intervals. It has three parts: a detector (weighing 5.6 kg), an external probe (weighing 0.65 kg), and a PC (only for data processing). The second spectrometer, a portable NIRS, uses only near-infrared wavelengths (950–1650 nm) in consecutive intervals of 2 nm. This is a very compact portable instrument (weighing 2 kg) containing a tablet, a probe, a detector, and a power battery in a single pistol-shaped unit, and it is therefore a very convenient tool to carry and rapidly collect spectra without need of a power source or PC. The third instrument, the cheapest of the three, is a miniaturized industrial spectrometer (Micro-NIRS), a hand-held instrument (weighing only 60 g) covering a similar range to the NIRS (905–1649 nm) but in intervals of 6 nm and only needing a PC/tablet as a power source for spectra collection.

### 2.3. Acquisition of Infrared Spectra of Meat

After seven days of aging, but no further processing of the meat, spectra were taken on a freshly cut, cross-sectional surface of the *Longissimus thoracis* muscle after 1-h exposure to the air at room temperature (23 ± 2 °C). This procedure mimics commercial practice where an aged carcass side is divided into the two quarters or from which individual beef steaks are obtained. Three replicate spectra were taken in reflection mode from different positions on the rib eye surface. All the spectra were taken at an angle of 90° to the cross-sectional surface of the meat. Prior to collecting the meat spectra, the spectrometers were calibrated using a standard white barium sulfate surface.

### 2.4. Analysis of the Minerals in the Meat

Mineral content was determined on representative subsamples of ground, freeze-dried meat, using the methods described in detail by Patel et al. [20]. Briefly, for each sample, 0.30–0.35 g of freeze-dried tissue was weighed and placed in a TFM vessel with 2 mL of 30% hydrogen peroxide and 7 mL of concentrated (65%) nitric acid, both of Suprapur quality (Merck Chemicals GmbH, Darmstadt, Germany). The prepared samples were subjected to microwave digestion (Ethos 1600 Milestone S.r.l. Sorisole, BG, Italy) as follows: Step 1, 25–200 °C in 15 min at 1200 W with P max 100 bar; Step 2, 200 °C for 15 min at 1200 W with P max 100 bar; and Step 3, 200–110 °C in 15 min. After cooling to room temperature, the dissolved sample was diluted with ultrapure water (resistivity 18.2 M Ω cm at 25 °C) to a final volume of 25 mL.

The mineral contents were determined with a Spectro Arcos EOP inductively coupled plasma–optical emission spectrometry (ICP-OES) (Spectro Analytical Instruments GmbH, Kleve, Germany). All instrument operating parameters were optimized for a 30% nitric acid solution as follows: axial plasma observation, crossflow nebulizer, Scott double-pass spray chamber, 3.0 mm inner diameter quartz torch injector, 1400 W plasma power, coolant gas 12.0 L/min, auxiliary gas 0.8 L/min, nebulizer gas 0.90 L/min, additional gas 0.20 L/min, sample uptake rate 2.0 mL/min, replicate read time 28 s, replicates 3, and pre-flush time 60 s. The three replicates from each meat sample were averaged before statistical analyses.

### 2.5. Validation of the Mineral Content of Meat

Calibration standards were prepared using multi- and single-element standard solutions (Inorganic Ventures Inc. Christiansburg, VA, USA) in 30% Suprapur nitric acid (Merck Chemicals GmbH, Darmstadt, Germany) to obtain similar matrices to the samples [20]. Concentrations of 0, 0.005, 0.02, 0.05, 0.2, 0.5, 2, and 5 mg/L of the analytes were prepared. We used the same concentrations of the calibration solutions for calcium, potassium, magnesium, sodium, phosphorous, and sulfur as for the other analytes, plus further concentrations of each of 20, 50, and 200 mg/L. The accuracy and precision of both methods were assessed with a blank solution, a low-level control solution (recovery limits ±30%), a medium-level control solution (recovery limits ±10%), and an international standard reference material NIST SRM 1577c (National Institute of Standards & Technology (NIST), Gaithersburg, MD, USA) prepared as described above. The measured and certified values were in excellent agreement for all the elements.

### 2.6. Editing the Mineral Data and Meat Spectra

The mineral contents of the meat samples were processed in two steps to identify outlier animals (first step) and to identify outlier samples within animals (second step). In the first step, the data were first analyzed with a mixed model that included breed and sex as fixed effects, and herd, animal within herd, and carcass side (residual) within animal as random effects [19]. Animal standard deviations (ASD) for mineral contents were calculated, resulting in two animals outside the interval mean ± 3 ASD being considered outliers and excluded from further analyses, for all minerals, leaving a maximum of 178 samples (89 animals × 2 carcass sides) for analysis. After excluding these two outlier animals, the second step involved calculating the residuals of the mineral contents with the previously-described model; individual samples outside the interval mean ± 3 RSD (residual standard deviation) were considered sample outliers for a given mineral and excluded from further analyses, as the samples were discarded with values before the error threshold of the mineral considered (16 out of 20 minerals had none of the samples excluded).

Outlier spectra were searched using the absorbance values (calculated by applying log (1/R) to the meat reflection data) extracted from the three instruments. Each spectrum was centered and standardized, and then Mahalanobis distances were calculated but no spectra showed a distance greater than the mean ± 3 SD and were considered outliers.

### 2.7. Mathematical Pretreatments of Meat Spectra

To better compare the three technologies/instruments, and avoid confounding factors due to differences in their data processing methods, the meat spectra were analyzed using the same methods in the R environment using PLS and ML-Matrix packages [21]. Meat spectra were preprocessed because pre-processing of spectral data is the most important step before chemometric bi-linear modeling [22]. A first dataset was obtained by spectra standardization and mean-centering (S/MC). A second dataset was obtained from the first using a standard normal variate (SNV), the Savitzky–Golay first- and second-order derivatives (SG) using a window size of 15 points with a third-order polynomial, after which the spectra were mean-centered (S/MC-SNV-SG). Lastly, a third dataset (S/MC-SNV-DT) was obtained from the first using an SNV followed by detrending (DT). However, we ultimately selected and report here in the tables only the results obtained from the S/MC-SNV-DT pretreatment, comparing the pre-treatments only in some figures. Mineral reference data from the laboratory were also standardized and mean-centered. All these steps were performed separately and independently for the three different instruments.

### 2.8. Infrared Spectra Calibration and Validation for Predicting the Mineral Content of Meat

Prediction models for the minerals were developed using partial least squares regression analysis (PLSR). In general, we used a maximum of 10 latent variables to build the PLSR calibration models for the various minerals. The prediction models were generated from a calibration dataset comprising 14 of the herds, while a validation dataset, comprised of the other herd, was used to externally validate the models and quantify their prediction ability. This process was repeated 15 times across the 15 herds using a “leave-one-herd-out validation” (LOHOV) strategy. The calibration models were externally validated by comparing the predictions obtained for the herds excluded from the calibration with the corresponding laboratory measured data. In this way, no samples from the same herd and sampling date were included in the calibration and validation datasets at the same time. Model performance was measured using the calibration coefficient of determination (*R*^2^_CAL_) and the external validation coefficient of determination (*R*^2^_VAL_). We also calculated and report here the root mean squared error of external validation (RMSE_VAL_).

### 2.9. Comparison of the Instruments

The accuracy of the three instruments according to the ISO (ISO 5725-1:1994) involves a combination of random components and a common systematic error or bias component. The performances of the instruments were evaluated by comparing the differences between the predicted and measured contents of each individual mineral in each sample. Trueness was measured using a mixed model that included the fixed effect of the spectrometer and the random effect of animal. Differences among the least squares means (LSM) of the three spectrometers were assumed to be differences in their trueness, while a significant difference between the LSM of an instrument and the expected 0.00 value was considered an indication of a biased result (over- or under-prediction according to the sign of the LSM).

Precision was measured by Levene’s test (the residuals were analyzed with the same mixed model but in absolute values). Levene’s test was done because it robustly tests for equality of variance [23]. Significant differences among the LSMs of the three spectrometers were assumed to be differences in the “average” residuals, i.e., the residuals from the different spectrometers are of different sizes, and the most precise is the one with the lowest LSM.

## 3. Results and Discussion

### 3.1. Detailed Mineral Profile of Beef

Twenty different minerals were detected in the beef samples: six essential macro-minerals (Na, Mg, P, S, K, and Ca), five essential micro-minerals (Cr, Mn, Fe, Cu, and Zn), and nine environmental micro-minerals (Li, B, Al, Ni, Sr, Sn, Ba, Ti, and Pb). Descriptive statistics of the concentrations of these 20 minerals are presented in Table 3. In general, the average contents found here fall within the wide range of results published in the scientific literature and were discussed in detail along with the effects of farm, breed, sex, age, weight, and carcass side in our previous study [19]. All these factors play a role in increasing the variability in the mineral concentrations of the meat samples, and hence also in building a robust prediction model. They were not, however, included in the calibration models in order that these would mimic the possible use of portable spectrometers in practical conditions.

### 3.2. Visible and NIR Spectra of Beef

The spectra of the beef samples taken with the three instruments are presented in Figure 1. As can be seen, the Vis-NIRS has a spectrum interval about double that of the NIRS and Micro-NIRS instruments. Whereas all three spectrometers cover all the area of the second overtone, and part of third overtone, only the Vis-NIRS covers the visible region of the spectrum, almost all the area of the third overtone, and the last part of the first overtone. Overall, the spectrum shows the typical spectral signature of meat tissue, characterized mainly by the protein, lipid, water, and pigment vibrational signal areas. The pattern of our beef spectra was generally similar to that previously reported for beef samples [13,24,25].

### 3.3. Models for Predicting the Mineral Content of Meat

A summary of the results of the calibration and external validation models (the latter obtained from herds and sampling dates not included in the calibration dataset) built using PLS regression for all three instruments is presented in Table 3. There are several other recent studies on the accuracy of predicting meat characteristics using NIR spectroscopy [26,27]. The best results were obtained predicting the chemical composition of organic matter of meat and the color traits, whereas modest results were obtained for the other physical traits, such as drip and cooking losses and meat tenderness [13,28,29,30]. Although prediction of mineral content is mainly correlative in nature, several studies on NIRS predictions of minerals have been published in the field of food and agriculture [3,31]. NIRS, especially portable ones, have rarely been used to predict the major minerals in meat.

It is worth noting that *R*^2^_VAL_ is always lower than *R*^2^_CAL_, sometimes dramatically so, especially in the case of the minerals with the lowest prediction accuracies, as can be clearly seen from the average values summarized in Figure 2 (top). As previously mentioned, mineral compounds are not or weakly associated with any specific bands in the infrared part of the electromagnetic radiation absorption spectrum [3], thus their prediction is substantially correlative in nature. This means that the combination of signals identified by PLSR for each mineral are highly specific to the batch of samples used (farm and beef system of origin, slaughter date and procedure, etc.). Several studies, especially the earlier ones, used a relatively small number of samples, often from a single batch (of, for example, animals used in an experiment) without cross-validation or external validation [3]. In such conditions, the reported *R*^2^ could also be high. The use of samples from different batches (different animals from different farms/areas/feeding systems and slaughtered and sampled on different dates), as in this study, introduces sources of variation that cannot always be captured by the absorbance spectrum, which reduces the accuracy of the statistics obtained from truly external validation but increases the robustness of the calibrations, thus mimicking the conditions found in practice [13].

Moreover, the maximum expected accuracy for predicting a trait by NIRS cannot be 100%, as it cannot be greater than the combined effect of the laboratory repeatability of the gold standard methods used for obtaining reference values and of the spectrometric repeatability of each wavelength absorbance by the analyzed matrix. These issues have been discussed in recent studies on the prediction of cheese composition [32] and beef quality traits [13]. Meat is a heterogeneous matrix due to the different distributions of muscle, connective, fat, and vascular tissues, and this can affect sample homogeneity for laboratory analyses as well as the variability among spectra collected in contiguous areas of the same meat section. Savoia et al. [13] clearly showed that the repeatability of absorbance by beef is highly dependent on individual wavelengths: it appears to be high in the range 500–1400 nm, but drops in the visible section of violet, and especially in the NIR > 1400 nm. This variability affects the usefulness of these wavelengths for predicting minerals (and other traits) and influences the maximum accuracy of predictions differently for different minerals (and other traits) according to the contribution of the different wavelengths to the prediction equations.

To make the results of the NIRS predictions of the minerals clearer, in the discussion of each mineral, we also refer to results obtained from the dataset of reference values (mineral contents determined by the gold standard ICP-OES method) reported in previous studies within the same project dealing with the effects of the main sources of variation. The content of minerals in meat was, in fact, related to the effects of the carcass batch (defined as the combination of the farm of origin and date of slaughter), the breed, and sex of animals, the individual animal within batch, and the carcass side within carcass (as both carcass sides were sampled, this variability was assumed to be the residual variability) [20]. In addition, the animal repeatability of each mineral analyzed was calculated. Other information from the same source adduced in our discussion concerns the covariance among the beef contents of the 20 minerals analyzed, and particularly the five latent explanatory factors explaining the greater part of all covariation. Moreover, to interpret the prediction performances for several minerals, probably derived more from indirect correlations with other traits than from direct vibrational signals, we also cite evidence relating to the correlations of the mineral contents with beef chemical composition, color traits, pH, cooking losses, and shear force illustrated in another study [19]. To make the text more readable, these two previous studies are not referenced again in the following paragraphs.

### 3.4. Predicting the Essential Macro-Mineral Content of Meat

P was the best predicted macro-mineral (Table 3) with all instruments, having the highest determination coefficients for calibration (*R*^2^_CAL_: 0.70–0.84) and external validation (*R*^2^_VAL_: 0.52–0.78), and an RMSE_VAL_ in the range of 68–103 mg/kg of fresh meat (i.e., 4–7% of the average content). It is worth noting that the P content measured on these same beef samples with ICP-OES had the highest animal repeatability (>90%) of all the macro minerals, meaning that the effect of sampling side of carcass, and all the variability due to sample processing and analysis, were very low for this element. The P content of meat was not affected by breed and sex of animals and its overall variability was minimally affected by individual animals within batch and carcass side within animal. The batch of carcasses, i.e., the farm of origin and the date of slaughtering, represented about 90% of all variability of P content in meat samples. As no information related to animals, farms, and dates is included in the prediction equation, NIRS prediction of this mineral is able to capture and account for the effects of different beef systems and feeding regime important for this mineral.

NIRS prediction of minerals is generally based on the presence of minerals in functional groups of organic matter. Therefore, the high predictability of P in beef could be due to its association with lipids (phospholipids), or to the phosphorylation process after slaughter of the animals. However, throughout the course of this project, we did not find any association between the P content and chemical composition (dry matter, protein, lipid, ash, cholesterol, and pH) of the beef samples. However, P could also be indirectly detected through its linkage with other organic complexes, chelates, and pigments. As Li et al. [33] pointed out, the phosphorylation process changes the meat color and pH. In support of this conjecture, we found a highly significant correlation in these same beef samples between the lab-measured P content and the redness, yellowness, chroma, and hue color indices of the cross-sectional surface of the meat.

Other macro minerals, such as Na, Mg, and S, showed poor to moderate predictability for all three instruments (*R*^2^_VAL_: 0.24–0.51, 0.26–0.58, and 0.28–0.55; respectively), while K and Ca had the lowest prediction accuracy with all the instruments, revealing no practical interests of these predictions. These results were in agreement with those obtained for animal repeatability from laboratory measurements, which were >60% for the first three elements and <50% for the other two. In fact, it should be kept in mind that expected maximum accuracy of any NIRS prediction is not 100%, but also it cannot be greater than the repeatability of analysis. In this case, it is animal repeatability, which includes not only instrumental repeatability but also sampling and sample processing repeatability. Differently from P, it was shown that sample/side variability accounted for more than 50% of overall variability in the case of K and Ca content of meat and that the batch represented about 25% of total variability for the former and only 5% for the latter mineral. Sampling both sides of each carcass in the same anatomical position turned out to yield useful information for discriminating between the effect of meat sampling and the effect of laboratory repeatability, and for interpreting the results relating to the effectiveness of NIRS prediction. Moreover, it can be seen in Table 3 that, moving from calibration to external validation, P exhibited a smaller decrease in *R*^2^ (−0.06 to −0.18 according to the different NIR spectrometers) than Na (−0.15 to −0.29), Mg (−0.15 to −0.29), S (−0.12 to −0.23), and especially K (−0.23 to −0.31) and Ca (−0.16 to −0.31). These differences could also be interpreted as a change in the ranking of the minerals moving from predictions based on direct oscillatory effects to indirect predictions, mainly correlative in nature. Here, too, the experimental design comprising 15 different batches of samples from different breeds and farms, and of different sexes and slaughter dates, allowed us to identify the level of statistical overfitting of the predictions if used in practice at the population level without proper external validation [13].

Viljoen et al. [15] reported a similar *R*^2^ to ours for the prediction of P (0.77), and an *R*^2^ greater than ours for Mg (0.84) in minced mutton samples. Kadim et al. [16] also reported a similar *R*^2^ for P (0.90) and a greater *R*^2^ for Ca (0.91) in chicken carcasses. González-Martín et al. [14] obtained much greater accuracies for predicting Ca, Na and K in Iberian pork (*R*^2^: 0.76, 0.78, and 0.64, respectively), although it should be pointed out that their results came from a very small number of meat samples (N = 34 for calibration), and that the *R*^2^ values they reported refer to a calibration with no internal or external cross-validation. Moreover, only eight samples of the same batch were retained for validation, although the statistics are not reported.

Prediction of Mg could be related to its role in lipoprotein metabolism and its relation to fatty acid and protein synthesis. Mg is also a co-factor of several enzymes [34], and it has been shown to have a positive correlation with high-density lipids [35]. Our analyses do not provide confirmation of significant correlations with beef chemical composition (although the correlation between Mg and cholesterol content at the farm level was +0.32, n.s.). As with P, we found associations between Mg content and beef color traits, and these last traits are known to be predicted by NIRS with good accuracy [12,13,26].

Very few studies have been carried out on the prediction of Na in meat. This element could be related to water content, which does not vary much in fresh *Longissimus thoracis* muscle samples from cattle. We found no correlation between Na content and any of the beef composition or quality traits in these samples, with the exception of some color indices. The prediction of S could be mainly related to sulfur-containing amino acids in meat proteins [35]. In these same samples, the S content was positively related to beef protein content and to color and negatively to cooking losses. The prediction accuracies for K and Ca were almost null, although they were correlated with the chemical composition of beef, a finding that may have to do with their low electronegativity, which leads to the formation of very weak hydrogen bonds that are difficult to detect with NIRS. In these cases, too, some of the studies on different agricultural and plant materials have found better predictabilities than those reported here, especially with longer NIR wavelengths, which, as already mentioned, are limited in the instruments used in this study and exhibit low animal repeatability in beef.

### 3.5. Predicting the Essential Micro-Mineral Content of Meat

The results of the calibration (obtained through internal random cross-validation) and external validation of the essential micro-minerals are presented in Table 3. These show that the iron content of meat had the highest *R*^2^_CAL_ (0.85–0.88) and *R*^2^_VAL_ (0.81–0.85), as well as an RMSE of 1041–1165 µg/kg. In the case of Zn, the *R*^2^_VAL_ (0.27–0.36) was much lower than the *R*^2^_CAL_ (0.55–0.62). This large difference confirms that the prediction of Zn is more correlative in nature and thus depends on the specific group of samples used for calibration. New samples sharing the same origin (same farms and dates of sampling of those used for calibration) could be predicted with acceptable accuracy, whereas samples totally independent from those used for calibration can be predicted with a less satisfactory accuracy. The other essential micro-minerals (Cr, Mn, and Cu) had lower *R*^2^_CAL_ (0.24–0.50) and almost null *R*^2^_VAL_ values (0.00–0.11). Thus, the interest of meat industry for their NIRS prediction in practical conditions is almost inexistent. A strong relationship between animal repeatability of the reference values and *R*^2^_VAL_ was also evident for the essential micro-minerals. The highest animal repeatability was for meat Fe content (>90%) and the lowest for Cr (20%). This means that, at least in part, the scarce performance of predictions based on NIRS is due not only to the modest link between the content of the mineral and the vibrational properties of the minerals, but also to the great influence that the variability meat sampling and processing and analytical procedure applied exerts on the content of the mineral in the samples analyzed.

There have been very few studies of NIRS predictions of essential micro-minerals in meat, and they are limited to just a few elements, such as Fe, Zn and Cu. Hong and Yasumoto [36] dealt with prediction of the iron content of various meats (beef, pork, rabbit, duck, horse, lamb, and mutton) using NIR in the range of 1100–2500 nm. They predicted total iron content with an *R*^2^ of 0.90, which is close to our results for Fe content. Viljoen et al. [15] obtained a similar cross-validation *R*^2^ for the prediction of Fe (0.77) in minced mutton samples, while González-Martín et al. [14] obtained an *R*^2^ of 0.84 for the prediction of Fe in pork meat.

The high Fe prediction accuracy can be explained by the fact that iron-binding proteins, heme iron (myoglobin, hemoglobin, and cytochromes), and non-heme compounds, ferritin and hemosiderin, are among the dominant chromophores in muscle [36], and that chromophores such as oxyhemoglobin (HbO_2_) and deoxyhemoglobin (Hb) are dominant in the visible range [37]. It is worth noting that Fe was positively correlated with dry matter and protein conten in our beef samples t. Beyond visible light, NIR range absorption at 1474 nm, corresponding to the first overtone of NH stretching vibration in peptide, and at 1574 nm, corresponding to the first overtone of NH stretching vibration, are related to the prediction of iron content [36].

Zn had the second highest prediction accuracy after iron with all three instruments. Schmitt et al. [3] concluded from their review that Zn is predictable in several food matrices. González-Martín et al. [14] predicted Zn in pork meat with a calibration *R*^2^ of 0.69, which is similar to our *R*^2^_CAL_ (and much greater than our *R*^2^_VAL_). It is worth noting that Zn exhibited high (80%) animal repeatability, and, unlike Fe, it had little relation to the content of the other minerals. Zinc characterized the fifth latent explanatory factor, substantially a one-trait factor, which explained 10% of total variance in the mineral profile. It is also worth noting that Zn was associated with meat composition (dry matter, protein, and fat), color traits, and cooking losses.

All the other essential micro-minerals were found to have poor predictability, hence no practical interest for meat industry. Aside from their very low animal repeatability, this could be due to fact that these minerals did not have any strong associations with beef chemical composition and quality traits.

### 3.6. Predicting the Environmental Trace Mineral Content of Meat

Calibration and external validation of the environmental micro-minerals are presented in Table 3. Pb had the highest *R*^2^_CAL_ (0.56–0.70) of all these minerals, but a much lower *R*^2^_VAL_ (0.20–0.40). In general, with the exception of Pb, the content of environmental trace minerals in meat are characterized by a poor predictability. These results confirm the theory that NIRS prediction of minerals depends especially on their links with organic molecules present in the tissues. These are often found for minerals having a metabolic/regulatory function in the animal body (essential minerals), whereas this does not happen for minerals resulting from environment/feed contamination, not having specific roles in the organism. To explain this calibration performance, it should be pointed out that the Pb reference values appeared from the previous studies to have high (80%) animal repeatability, while Pb tended to be negatively correlated with all the other minerals except B. This is the only mineral, aside from Fe, to be explained by two latent factors: Pb is associated with most of the other elements in the first “quantitative” factor and, together with B, is associated negatively with K in the third latent factor (explaining 16% of total mineral covariance). We may therefore hypothesize that the prediction of Pb is affected by signals specific to many other minerals.

Few studies have dealt with prediction of these environmental minerals using NIRS. Some have reported predictions of Pb in foods other than meat, probably as a result of its linkage with lipids. However, the *Longissimus thoracis* muscle has a modest fat content, and in our study the Pb content was not correlated with lipids, but it was negatively correlated with protein, ash, and cholesterol contents. Correlations with color and meat quality traits were not significant, except for the positive correlation with cooking losses.

Ti showed a much lower prediction accuracy (*R*^2^_CAL_ 0.45–0.50; *R*^2^_VAL_ 0.13–0.25) than Pb. In our previous study, Sn had good animal repeatability, also because it exhibited a very high herd/date effect. This fully confirms the strong “environmental” nature of this trace element. It is worth noting that we observed no correlation between Sn and beef composition and quality traits. Only beef color was (negatively) correlated with its Sn content. However, the (modest) predictability of Sn may be due to the fact that Sn, similar to Pb, was mainly negatively correlated with the other minerals, and this is confirmed by its inclusion in the first “quantitative” latent factor, although it is the only element having negative weight compared with the others.

All the other environmental minerals (Li, B, Al, Ti, Ni, Sr, and Ba) were not able to be predicted. This was mainly associated with low animal repeatability (<70%), with the exception of Li and B, which had high animal repeatability (>90%), greater than that of Pb and Sn. Their covariation with other minerals was generally low with the exception of Al, Ti, and Ba, which were associated with some macro-minerals (Mg, P, and S) and Cr in the “quantitative” latent explanatory factor, and of B, which was negatively correlated with K. Some correlations, often negative, were found between the chemical composition of the beef samples and Li, B, Ni, and Sr; between color traits and Li, Al, and Ti; (unfavorably) between pH and Sr; and between cooking losses and B.

### 3.7. Comparison of the Prediction Abilities of the Different NIR Spectrometers

The ranking of the near infrared spectrometers was very different according to whether it was based on calibration (*R*^2^_CAL_, obtained using internal cross-validation) or on external validation (R^2^_VAL_, obtained with the leave-one farm/date-out strategy). Looking first at the calibration results, and considering instruments whose *R*^2^ differed by less than 0.05 as arbitrarily equivalent, Table 3 shows that the transportable Vis-NIRS ranked first for 17 of the 20 minerals analyzed (which included all the micro-minerals), and second for three; it did not rank last for any of the minerals. The portable NIRS was first only in the four cases in which there were no relevant differences between the instruments and was last 16 times. Finally, the hand-held Micro-NIRS ranked first 11 times and last 9 times. The superiority of the Vis-NIRS over the NIRS and Micro-NIRS instruments for calibration could, in view of the technical features summarized in Table 2, be attributed to the wider wavelength range tested (350–1830 nm vs. 950–1650 nm with NIRS and 905–1649 nm with Micro-NIRS), and its data-point interval (1 nm vs. 2 and 6 nm, respectively), and consequently the number of wavelengths included in the spectra (1481 vs. 351 and 125, respectively). With regard to the differences between the NIRS and Micro-NIRS, the causes should be sought in technical features other than spectrum range (similar) and data-point interval and number (in favor of NIRS). The superiority of Vis-NIRS by virtue of the number of data points detected was particularly evident when the S/MC and SNV pretreatments of spectra were followed by SG, as is clear in Figure 2 (top). An instrument yielding a large number of dependent variables (absorbance values) is more effective in exploiting the large matrix of covariances existing among the spectrum data-points. This large matrix could be used for predicting the mineral contents of samples coming from the calibration dataset or from different samples having the same origin (animals reared in the same farms and slaughtered in the same dates).

Regarding external validation, when samples predicted were totally independent from those used for calibrations (from animals reared in different farms and slaughtered in different dates), it can be seen in Table 3 that 9 of the 20 minerals analyzed (Ca, Cr, Mn, Li, B, Al, Ni, Sr, and Ba) had negligible predictability (*R*^2^_VAL_ ≤ 0.10) with all three spectrometers tested. This means that the scarce prediction performances are not due to any technical aspect of the instruments and number of data-points collected, but rather to the specificities of these minerals. These minerals are probably not linked to organic molecules present in the meat and predictable by NIRS and are not correlated to other physical traits that can be predicted by NIRS, e.g., many color traits.

Of the other 11 minerals, Vis-NIRS ranked first in three cases, second in six cases, and last in one case; NIRS ranked first in three cases and last in the other eight cases; and Micro-NIRS ranked first for ten minerals and second for one. In the case of external validation, it seems clear that spectrum range and data-point interval and number, so important for internal calibration, do not play any important role in determining the accuracy of prediction. The smallest spectrometer (Micro-NIRS) predicted a greater number of minerals (4–5 out of 20, according to spectra pretreatment) with acceptable accuracy (*R*^2^_VAL_ ≥ 0.49, i.e., r ≥ 0.70) than the other two larger instruments (1–3 minerals for both NIRS and Vis-NIRS), as shown in Figure 2 (bottom). It is worth noting that the effect of mathematical pretreatment, which is very large in the case of calibration (especially for Vis-NIRS), is negligible on *R*^2^_VAL_ (Figure 2).

The statistical analyses of the errors of predictions (the difference between the mineral content of a beef sample measured with the gold standard method and that predicted with the spectrometric method) revealed that none of the instruments yielded biased results (trueness of prediction) for any mineral, because the average of the errors never differed statistically from the expected 0.00 value (Table 3). On the other hand, there were significant differences in precision for three minerals (Na, Mg, and P; Table 3), because the absolute value of errors of prediction was lower with Micro-NIRS than with NIRS (Figure 3).

This reversal of the instrument ranking has to do with the entity of the decrease in *R*^2^ moving from calibration (internal cross-validation) to external validation, which was −0.29 ± 0.10 for Vis-NIRS, −0.23 ± 0.08 for NIRS, and −0.20 ± 0.08 for Micro-NIRS, using the S/MC-SNV-DT spectra pretreatments. In their predictions of beef quality traits, Savoia et al. [13] also observed a much greater decrease in *R*^2^ moving from calibration to external validation in the case of Vis-NIRS than in the case of Micro-NIRS. In a different context—the authentication of dairy systems directly from milk and cheese characteristics using five alternative sources of information (FTIR spectrum and fatty acid profiles of milk, NIR spectrum, flavor fingerprinting, and sensory description of cheeses)—we found that, when sources of information were evaluated on internal calibration, they were more effective when they were characterized by a larger number of data-points per sample. However, these same methods were also those exhibiting the largest decreases in effectiveness moving to external validation [38]. As a result, we should consider with caution the results of several studies based on internal calibration or random cross-validation, in which the calibration and validation dataset include different samples with the same origin (batch, farm, sampling date, etc.), as their effectiveness statistics could be overoptimistic due to the effect of relationships specific to the given database and not extendable to new samples/batches/farms/sampling dates, etc. In the case of traits whose predictions are correlative in nature, such as for many minerals, only external validation can really guarantee that the estimated effectiveness statistics reflect the conditions observed in practice, as clearly demonstrated by Wang and Bovenhuis [39].

Furthermore, the usefulness of extending the NIR spectrum to the visible spectrum is questionable, as it appears from the results in Table 3 that in terms of *R*^2^_VAL_ Vis-NIRS does not seem to be any better than NIRS and Micro-NIRS for predicting the minerals more closely correlated with beef color traits, e.g., Fe, Zn, P, and S [19]. In a large survey comparing the prediction of beef color traits using NIR spectrometry in the abattoir with their corresponding laboratory measures [13], the Micro-NIRS predictions for color traits displayed the same level of accuracy as Vis-NIRS. In a study on a different food—cheese—using different instruments (transportable Vis-NIRS vs. two bench-top NIRS), Stocco et al. [32] compared validation performances when the entire spectrum of Vis-NIRS or only one of three portions of the spectrum is used. In the prediction of cheese lightness (L*), the results from the entire spectrum were similar to those from only one of the three portions of the spectrum (or with the bench-top NIRS instruments, when not analyzing the visible light). In contrast, when estimating the other color traits (a*, b*, C*, and H*), the results from the entire Vis-NIR spectrum or just the visible light fraction were greatly superior to those from the NIR fractions.

Moreover, most of the minerals were detected in the upper NIR wavelength range, where the Vis-NIR instrument has greater coverage, but where beef samples have low repeatability [13].

Figure 4 plots the accuracy (*R*^2^_VAL_) of the predictions of each mineral (Table 3) according to its average content in the beef samples analyzed (Table 3), and according to the instrument used to acquire the spectra. First, we can see that the wide variation in the results depends very little on the average concentrations of the minerals in the meat samples. We can also see that the differences in terms of accuracy of prediction among the different minerals (including those with similar concentrations in the meat) are much greater than the differences among the three instruments for each mineral. Moreover, the entity of the differences among the three instruments also seems to be independent of the concentrations of the minerals in the meat samples. Therefore, we can conclude that the accuracy of the predictions of all these instruments seems to be unaffected by the concentrations of the minerals in beef.

Given that the repeatability and reproducibility of the NIR absorbance spectra [13] is much greater than the reproducibility of the gold standard methods, we might speculate that the NIRS predictions, even where the *R*^2^_VAL_ is not very high, are able to capture much of the animal variability and could therefore be useful for monitoring quality. Studies carried out on beef quality traits [12,13], milk coagulation properties [40], cheese yield and milk nutrient recovery [41], enteric methane emissions of dairy cows [42], and the fatty acid profiles of bovine milk [43] and sheep’s milk [44], have shown that the genetic correlation between infrared predictions and gold standard values are generally greater than the phenotypic correlations. The authors have therefore proposed the use of infrared predictions at the population level for the genetic improvement of those traits that are difficult to measure. These findings are supported by the knowledge that many infrared wavelength absorbances of milk are heritable [45,46,47]. No information on the genetic background of beef absorbance spectra is available.

## 4. Conclusions

NIR spectroscopy cannot analyze the metals per se, but it is useful for detecting minerals associated to organic molecules. This study is the first to employ this rapid method to predict the contents of 20 essential macro-, essential micro-, and environmental micro-mineral elements in beef using portable and handheld NIR instruments. The results show that, among the 20 minerals, P and Fe were predicted with the greatest accuracy, taking into account the heterogeneity of the meat samples and repeatability of the gold standard method. The predictions of the other macro- (Na, Mg, and S) and micro-minerals (Zn and Pb) were less accurate, but they could still be useful in monitoring the production chain, pre-screening samples for further analyses, or perhaps for genetic selection purposes. Even though the three portable spectrometers used in this study were very different in terms of accuracy of calibration, the best being the Vis-NIRS, which acquires data from the widest and most clearly-defined spectrum, they have similar levels of precision in the external validation for 17 out of 20 minerals, while the smallest instrument (Micro-NIRS) exhibited greater precision for the Na, Mg, and P contents of beef. This study allowed us to gain a better understanding of the role of some of the technical features of the instruments and we showed that very simple, inexpensive, hand-held instruments can be used directly on the muscle surface in the real working conditions of abattoirs and meat processing plants, without the need to collect and process meat samples, with considerable efficacy.

## Figures and Tables

**Figure 1 foods-09-01389-f001:**
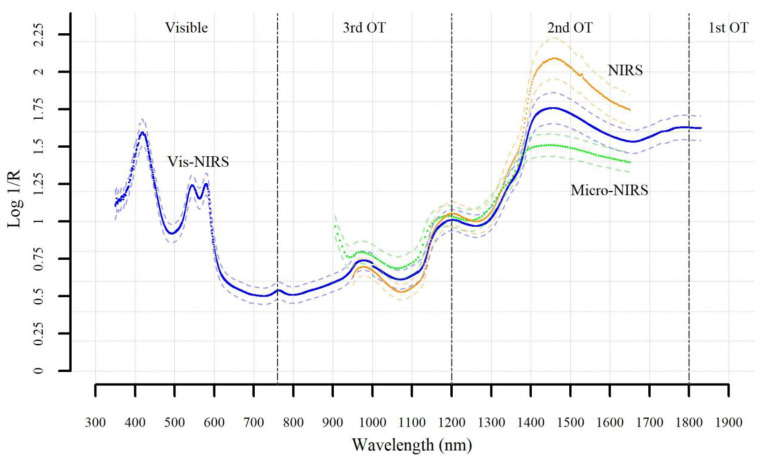
Mean absorbance spectra of 192 beef samples recorded by Vis-NIRS, NIRS, and Micro-NIRS according to wavelength. The spectra show approximately where the first, second, and third overtone (OT) vibrations occur, as well as the visible range.

**Figure 2 foods-09-01389-f002:**
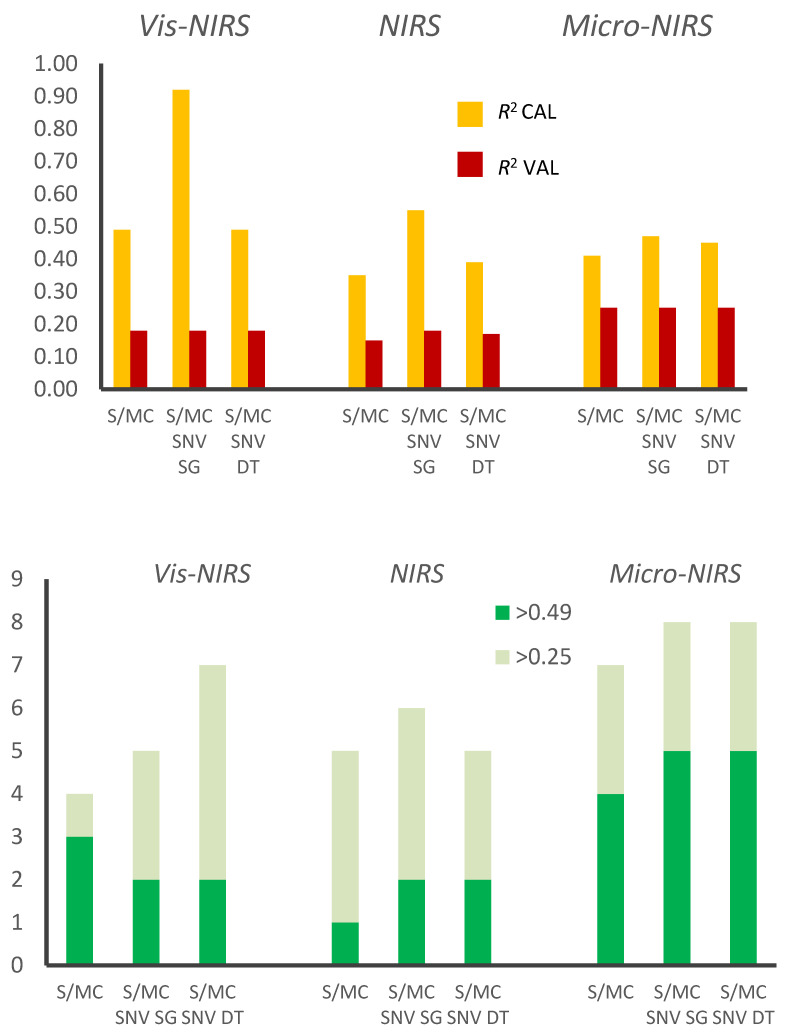
Average determination coefficients (top) of calibration (*R*^2^_CAL_) and external validation (*R*^2^_VAL_), and the number of minerals out of 20 (bottom) with *R*^2^_VAL_ > 0.49 (*r* > 0.70) and 0.25 (*r* > 0.50) obtained with the three infrared spectrometers and the three combinations of mathematical pretreatments of the absorbance values of the spectra. X-MC, spectra standardization and mean-centering; SNV, standard normal variate; SC, Savitzky–Golay first- and second-order derivatives with a third-order of polynomial; DT, detrending.

**Figure 3 foods-09-01389-f003:**
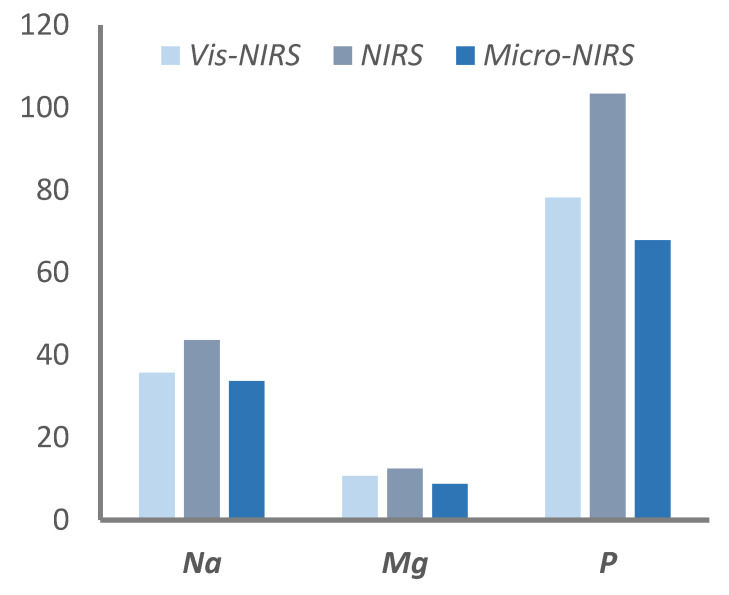
Average of absolute errors of prediction (in mg/kg of meat) obtained from external validation for the minerals presenting significantly different levels of precision with the three infrared spectrometers.

**Figure 4 foods-09-01389-f004:**
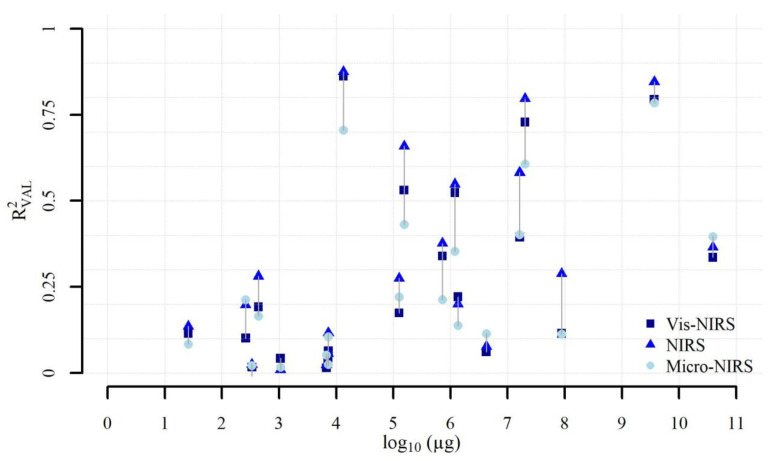
Relationships between the mean concentrations of the minerals in meat and the accuracy (*R*^2^_VAL_) of their predictions obtained from the three NIR spectrometers.

**Table 1 foods-09-01389-t001:** Experimental design.

	N ^1^	Type
*Animals and meat samples*		
Breeds	2	Chianina, Romagnola
Sex	2	Young bulls, Heifers
Farms/slaughter dates	15	-
Animals sampled	91	89 after outliers’ exclusion
Carcass sides/animal	2	Right side, Left side
Total rib-cut meat samples analyzed	178	2 × 89
*Minerals*		
Minerals analyzed/meat sample	20	Na, Mg, P, S, K, Ca, Cr, Mn, Fe, Cu, Zn, Li, B, Al, Ti, Ni, Sr, Sn, Ba, Pb
Total minerals data used	3404	20 × 178 = 3560 − 156 values below error threshold
*NIRS spectra*		
Spectrometers tested	3	Vis-NIRS, NIRS, Micro-NIRS
Replicates per spectrum	3	averaged by the instrument
Spectra obtained/meat sample/instrument	3–5	3: Vis-NIRS and Micro-NIRS, 5: NIRS
Total spectra stored	1958	(3 + 3 + 5) × 178
Absorbance values/instrument/spectrum	125–1481	(see Table 2)
Total absorbance values stored	1169,994	(178 × 3 × 1481) + (178 × 5 × 351) + (178 × 3 × 125)

^1^ total number.

**Table 2 foods-09-01389-t002:** Main characteristics of the portable NIR spectrometers used to predict the mineral content of meat.

	Vis-NIRS	NIRS	Micro-NIRS
*Instrument*			
Denomination	LabSpec 2500	Aurora NIR	Micro NIR Pro
Producer	ASD Inc.	Grain It	JDSU
Address	Boulder (CO)	Padova (PD)	San Jose (CA)
Country	USA	Italy	USA
*Characteristics*			
Type	transportable	portable	hand-held
Spectrometer size (cm)	13 × 37 × 29	23 × 12 × 7	4.5 × 4.4 × 4.0
Spectrometer weight (g)	5600	2000	60
Sample preparation	none	none	none
Method	reflectance	reflectance	reflectance
Operating temperature	0 + 40 °C	0 + 40 °C	−20 + 40 °C
Spectra storage	internal	internal	PC or tablet
Connectivity/interface	10/100 Base T Ethernet	2× USB, Ethernet	USB 2.0 (480 Mbps)
Power source	cable	internal battery	USB 2.0 (480 Mbps)
*Illumination*			
Source	halogen	halogen	2 vacuum tungsten lamps
Aperture	2.0 mm	2.5 mm	2.5 mm
Light detection	external probe	internal	internal
Probe connection	optical fiber	-	-
External probe size (cm)	26 × 10 × 5	-	-
External probe weight (g)	654	-	-
Detector type	Diode Array (Si,inGaAs)	InGaAs array	InGaAs array
Measurement time	0.1 s	0.5 s	0.5 s
Scanning method	external reference	internal reference	external reference
Computer/tablet	external PC	incorporated	external PC
*Spectrum*			
Waves range (nm)	350–1830	950–1650	905–1649
Data point interval (nm)	1	2	6
Data point per sample	1481	351	125
Replicates per spectrum	3	3	3
Spectra collected	3	5	3
Absorbance calculation	A = log(1/R)	A = log(1/R)	A = log(1/R)

**Table 3 foods-09-01389-t003:** Descriptive statistics and performances of the calibration and validation models for predicting mineral contents from meat absorbance spectra using three portable spectrometers.

Minerals	Descriptive Statistics			Vis-NIRS			NIRS			Micro-NIRS			Accuracy (*F*-Value)	
	Samples	Mean	SD	*R* ^2^ _CAL_	*R* ^2^ _VAL_	RMSE	*R* ^2^ _CAL_	*R* ^2^ _VAL_	RMSE	*R* ^2^ _CAL_	*R* ^2^ _VAL_	RMSE	Trueness	Precision
Essential macro, mg/kg														
Na	178	437	49	0.68	0.48	36	0.53	0.24	43	0.66	0.51	34	0.14	3.27 *
Mg	178	179	14	0.65	0.43	11	0.55	0.26	12	0.73	0.58	9	0.61	3.45 *
P	178	1490	145	0.80	0.71	78	0.70	0.52	103	0.84	0.78	68	0.36	4.64 **
S	178	1354	84	0.56	0.34	70	0.49	0.28	72	0.67	0.55	52	0.09	1.32
K	178	2816	110	0.35	0.04	126	0.28	0.05	114	0.43	0.19	74	0.32	1.25
Ca	178	49	19	0.31	0.00	22	0.19	0.03	22	0.20	0.00	20	0.28	0.27
Essential micro, µg/kg														
Cr	103	20	61	0.50	0.00	71	0.26	0.00	68	0.24	0.00	75	0.35	0.15
Mn	178	49	10	0.44	0.06	11	0.33	0.09	10	0.33	0.06	10	0.16	0.61
Fe	178	14,352	2530	0.88	0.83	1041	0.85	0.79	1165	0.88	0.81	1088	0.59	1.02
Cu	178	473	99	0.46	0.08	105	0.27	0.08	97	0.29	0.11	96	0.04	0.02
Zn	178	39,632	5456	0.62	0.27	4968	0.55	0.36	4420	0.56	0.33	4479	0.02	0.56
Environmental, µg/kg														
Li	178	4	2	0.41	0.08	3	0.35	0.00	2	0.35	0.02	2	0.18	0.14
B	178	171	120	0.43	0.07	127	0.43	0.06	118	0.43	0.10	112	0.05	0.32
Al	178	895	1200	0.32	0.00	1308	0.23	0.03	836	0.25	0.03	870	0.10	0.28
Ti	178	15	7	0.39	0.14	7	0.30	0.07	7	0.39	0.24	6	0.04	0.42
Ni	168	44	96	0.26	0.00	111	0.10	0.02	108	0.11	0.00	105	0.03	0.26
Sr	178	51	29	0.28	0.00	35	0.16	0.03	34	0.20	0.00	33	0.02	0.30
Sn	178	378	208	0.49	0.13	210	0.45	0.14	199	0.50	0.25	186	0.04	0.48
Ba	156	14	11	0.32	0.01	12	0.29	0.03	11	0.33	0.03	12	0.04	0.10
Pb	129	62	14	0.70	0.34	12	0.56	0.20	12	0.67	0.40	11	0.14	1.82

*R*^2^_CAL_= determination coefficient of calibration, *R*^2^_VAL_= determination coefficient of validation, RMSE= root mean square error. *: *P* < 0.05; **: *P* < 0.01.

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
