# Peer review of "Predicting the Content of 20 Minerals in Beef by Different Portable Near-Infrared (NIR) Spectrometers"

_foods, 2020, doi:10.3390/foods9101389_

Round 1
Reviewer 1 Report
This study explores difference NIRs devices in order to determine the main 20 metals on beef.
The introduction is correct and introduce the importance of the minerals determination on meat, the different techniques of the analysis for the minerals determination and some applications of the determinations of the minerals in meat samples.
Section 2.1 A figure could clarify the experimental design proposed for this study.
Line 105. Indicate the temperature of the room in order to improve the reproducibility of the study.
Line 130. The reference must be indicated acording to the authors guidelines.
Section 2.7 Some references must be added for the explanation of mathematical methods applied.
Section 3.2. For the readers could be interesting to show the spectra with and without preprocessing in the same figure.
Why the R2 values are so low, this explanation is key for the aceptation of the paper in the journal. Taking into account that the NIRs are not sensitive to metals. Additionally R2 < 0.25 indicates that no exists relationship between the predicted values obtained by PLS-R and the real measures obtained by traditional analytical methods. I suggests that only the metals with R2 > 0.25 should be the metals deeply analyzed.
Conclusions must be extended.
Author Response
REVIEWER 1:
This study explores difference NIRs devices in order to determine the main 20 metals on beef.
The introduction is correct and introduce the importance of the minerals determination on meat, the different techniques of the analysis for the minerals determination and some applications of the determinations of the minerals in meat samples.
AU: We thank the reviewer for his/her interest and time dedicated to the improvement of our paper. We really appreciated it.
Re: Section 2.1 A figure could clarify the experimental design proposed for this study.
AU: We found difficult to summarize the experimental design in a Figure, so we prepared a new Table (Table 1) for this aim. Table is included in manuscript. Changes can be seen in line 73-74 and table at line 88-90.
Re: Line 105. Indicate the temperature of the room in order to improve the reproducibility of the study.
AU: Done. Line 115
Re: Line 130. The reference must be indicated according to the authors’ guidelines.
AU: Line 139, Reference style is corrected
Re: Section 2.7 Some references must be added for the explanation of mathematical methods applied.
AU: Inserted a reference. Line 168
Re: Section 3.2. For the readers could be interesting to show the spectra with and without preprocessing in the same figure.
AU: The different pre-treatment cannot be included in the same picture because values obtained are completely changed respect to raw spectra and are completely different among them. In any case they depend directly from the raw spectra so the comparison among the raw spectra of the 3 instrument is the most informative.
Re: Why the R2 values are so low, this explanation is key for the acceptation of the paper in the journal. Taking into account that the NIRs are not sensitive to metals. Additionally R2 < 0.25 indicates that no exists relationship between the predicted values obtained by PLS-R and the real measures obtained by traditional analytical methods. I suggests that only the metals with R2 > 0.25 should be the metals deeply analyzed.
AU: Please consider that some published papers present only the calibration R2. In our work, with Vis-NIRS all minerals obtained calibration R2 > 0.25, as did 17 out of 20 with NIRS and 16 out of 20 with Micro-NIRS. Many papers reports R2 obtained with random cross-validation (in groups or through “leave one sample out” technique) and often these R2 are not much lower than calibration R2. An important result of our work is the demonstration that a really external validation could yield much lower R2 and then that published papers are sometimes over-optimistic. We fully agree that below 0.25 the practical value of prediction is almost null, and in fact, in the conclusions, we report that only 2 minerals (P and Fe) out of 20 had very good predictions and 5 others were less accurately predicted.
Re: Conclusions must be extended.
AU: Authors guide invite to keep conclusions very short.

Reviewer 2 Report
The authors presented the research article: “Predicting the content of 20 minerals in beef by different portable near-infrared (NIR) spectrometers”, regarding the comparison of three different portable NIR spectrometers by their ability of predicting various meat minerals though previously calibrated multivariate models. This could be of great interest for meat industry as well as of the audience of Foods journal, while the manuscript is well written discussing plenty of issues.
Revisions:
Line 48: Please put “2” at the end of sentence into brackets (reference).
Line 56: However, it would be great if in this study a fourth type of instrument (i.e. benchtop NIR spectrometers) could be included in the comparison, to see deviations from “professional” benchtop NIRS.
Line 112: If known, please add the particle size of the grinded sample.
Line 157: It would be more helpful for the audience to add also the R packages that used for the multivariate analysis.
Lines 174-175: This sounds great. However, are you sure that the necessary variance was included in the model and it is not just an artifact?
Line 187: Please provide a reference of the LEVENE’s test.
Line 202 (Table 2): I am missing the CV values. Please add them too to complete the table. Also, it would be useful to include the RMSE of all Cal, CV and Val.
Lines 225-227: That is true, but in these cases the model is not robust and you need to recalibrate with a different data set separation. Have you tried it?
Lines 234-239: That is also true (in connection to the previous comment) and related to the included variance during the calibration. Therefore, you could re-divide your samples in such way that you include as much variance as you can in your calibrated models in order to be even more robust with future unknown samples. Furthermore, chemometric models are dynamic, meaning that in practice you have to update them with more and more samples constantly in order to capture as you said this variance. One thing that you can do for now however, is to order your samples in an ascent order and choose every fifth (or whatever you think) to create your Val set. The rest will be used for the calibration.
Line 277: Why you refer to ICP-OES with the term gold standard method. I think it is better to use the exact name of the method, i.e. ICP-OES (counting also less words).
Line 292: An R2 value of 0.24 is not showing moderate but poor predictability. Maybe you should add the phrase poor to moderate predictability in this sentence.
Line 376: It could be this but it could be also what I said before regarding the sample set distribution. Did you also tried to reduce variable number, keeping only significant ones in order to enhance your predictive power? You apply a jack-knifing tool (such as Martens and Martens, 2000, algorithm) to recalibrate your model with only the significant variables. Furthermore, did you tried to apply different spectral transformations for this model?
Line 384: Please change “Tin” to “Ti”.
Line 392: I would suggest the phrase “null predictability” to “was not able to be predicted”.
Author Response
The authors presented the research article: “Predicting the content of 20 minerals in beef by different portable near-infrared (NIR) spectrometers”, regarding the comparison of three different portable NIR spectrometers by their ability of predicting various meat minerals though previously calibrated multivariate models. This could be of great interest for meat industry as well as of the audience of Foods journal, while the manuscript is well written discussing plenty of issues.
AU: We would like to thank you for your appreciation of our work and helping us to improve this by your suggestions.
Revisions:
Re: Line 48: Please put “2” at the end of sentence into brackets (reference).
AU: Corrected by inserting []. Line 48
Re: Line 56: However, it would be great if in this study a fourth type of instrument (i.e. benchtop NIR spectrometers) could be included in the comparison, to see deviations from “professional” benchtop NIRS.
AU: In this study we only used handheld and portable NIRS because our purpose was to use NIR spectroscopy directly on the carcasses and in real scenario in meat processing unit. This is why we did not use benchtop NIR instrument.
Re: Line 112: If known, please add the particle size of the grinded sample.
AU: We have followed the standard procedure of grinding and provided all the necessary operational parameters. However, we did not measure the particle size. Sorry.
Re: Line 157: It would be more helpful for the audience to add also the R packages that used for the multivariate analysis.
AU: Name of packages is provided. Line 167.
Re: Lines 174-175: This sounds great. However, are you sure that the necessary variance was included in the model and it is not just an artifact?
AU: We have used all the 14 farms at a time in calibration models and we repeated this procedure for 15 times. In this way we can be sure that all the necessary variation is included in the models.
Re: Line 187: Please provide a reference of the LEVENE’s test.
AU: Inserted a reference. Line 200.
Re: Line 202 (Table 2): I am missing the CV values. Please add them too to complete the table. Also, it would be useful to include the RMSE of all Cal, CV and Val.
AU: With CV do you means coefficient of variation or cross-validation? The first is derived directly from mean and SD values. The second was not included because of space limit. We did a random cross-validation, but this yields overoptimistic results because samples in validation sets are different from those in calibration but are not independent from them, coming from the same batches of samples (same farm, same slaughterhouse, same date of slaughter, same sample storage conditions, same date of mineral analysis, etc.) Also the RMSE of calibration was omitted because of space limits, but it could be easily obtained on the basis of the RMSE of val and the 2 R” (cal and val).
Re: Lines 225-227: That is true, but in these cases the model is not robust and you need to recalibrate with a different data set separation. Have you tried it?
AU: Please see next point.
Re: Lines 234-239: That is also true (in connection to the previous comment) and related to the included variance during the calibration. Therefore, you could re-divide your samples in such way that you include as much variance as you can in your calibrated models in order to be even more robust with future unknown samples. Furthermore, chemometric models are dynamic, meaning that in practice you have to update them with more and more samples constantly in order to capture as you said this variance. One thing that you can do for now however, is to order your samples in an ascent order and choose every fifth (or whatever you think) to create your Val set. The rest will be used for the calibration.
AU: we agree with reviewer that cross-validation yields higher (apparent) R2 and that a dynamic approach is optimal, but it is not so easy and cheap to continuously analyze the mineral content of new meat sample and to carry on new calibrations and validations. This is the reason why we decided to include in the paper the two extreme situations: the calibration results (the highest R2) and the really external validation results (the lowest R2). Researchers and laboratories know that they could obtain results intermediate if they are able to update frequently the calibration/validation datasets.
Re: Line 277: Why you refer to ICP-OES with the term gold standard method. I think it is better to use the exact name of the method, i.e. ICP-OES (counting also less words).
AU: Suggested word is inserted. Line 289.
Re: Line 292: An R2 value of 0.24 is not showing moderate but poor predictability. Maybe you should add the phrase poor to moderate predictability in this sentence.
AU: Done (please see also the answer to the similar objection of reviewer 1).
Re: Line 376: It could be this but it could be also what I said before regarding the sample set distribution. Did you also tried to reduce variable number, keeping only significant ones in order to enhance your predictive power? You apply a jack-knifing tool (such as Martens and Martens, 2000, algorithm) to recalibrate your model with only the significant variables. Furthermore, did you tried to apply different spectral transformations for this model?
AU: We have checked sampled set distribution, they have similar distribution in calibration and validation set. So, there could not be any reason for biased results. We did not try variables selection. However, we have used several combinations of preprocessing of spectra and we have reported the combination which yielded better results.
Re: Line 384: Please change “Tin” to “Ti”.
AU: Done. Line 398.
Re: Line 392: I would suggest the phrase “null predictability” to “was not able to be predicted”.
AU: Corrected as per your suggestion. Line 406.

Round 2
Reviewer 1 Report
The authors have improved the quality of the paper but some changes does not relevant for the reviewer. Consequently major changes must be realized in the paper.
The main drawback of this paper is the low values of R2 for the most of the minerals studied in this study. There are a lot of papers related with NIRs measures and quality and texture properties of the beef in the scientific literature that they show high R2 values with a external batch for the validation of the results. This lack must be deeply improved (Lines 286-512, Section 3.4, 3.5, 3.6 and 3.7). This point is the key for the acceptation of the paper.
Other minor changes:
- The room temperature must be indicated with the value ± standard deviation. e.g. 23 ± 2.
Author Response
Dear Reviewer,
Please find the corrections and explanation you have sought in order to accept the paper for publication. We would like to thank you for your insightful suggestions to make our article better. Thank you once again.
Corrections are below:
Re: The authors have improved the quality of the paper but some changes does not relevant for the reviewer. Consequently major changes must be realized in the paper.
Re: The main drawback of this paper is the low values of R2 for the most of the minerals studied in this study. There are a lot of papers related with NIRs measures and quality and texture properties of the beef in the scientific literature that they show high R2 values with a external batch for the validation of the results. This lack must be deeply improved (Lines 286-512, Section 3.4, 3.5, 3.6 and 3.7). This point is the key for the acceptation of the paper.
AU: We agree that we have obtained low R2 values for several minerals. But this result was expected.
In the introduction we wrote (L 42-46): “NIRS is a method that is sensitive to hetero atomic covalent bonds; therefore, it is not sensitive to metals. Minerals in fact are not or weakly associated with any specific bands in the infrared part of the electromagnetic radiation absorption spectrum, but the concentrations of some minerals can be predicted using proper chemometric treatment of the whole spectrum if they are bounded to some organic molecule”.
Consequently, in the conclusion we wrote (L 516-524): “Few studies have been carried out on the prediction of detailed mineral profile of beef using NIR spectroscopy because this method cannot analyze the metals per se, but are useful for detecting minerals associated to organic molecules. This study is therefore the first to employ this rapid method to predict the contents of 20 essential macro-, essential micro- and environmental micro-mineral elements in beef using NIR-based portable instruments. Among the 20 minerals, P and Fe were predicted with the greatest accuracy, taking into account the heterogeneity of the meat samples and repeatability of the gold standard method. The predictions of the other macro- (Na, Mg and S) and micro-minerals (Zn and Pb) were less accurate, but they could still be useful in monitoring the production chain, pre-screening samples for further analyses, or perhaps for genetic selection purposes”.
We also agree with you that there are several papers predicting some meat characteristic with external validation with high R2 values, but please consider that, as expected, they are relative mainly to composition of organic matter of meat (protein, fat, fatty acids, etc.) and to the meat color traits, whereas other physical characteristics of meat (drip losses, cooking losses, shear force, etc.) show modest prediction results, if external validation is carried out. So, a detailed comparison with studies on meat composition is not useful for the aims of this study, but we have shortly summarized these results citing 2 review papers and 3 articles not previously cited among those published in the last 3 years in the major journals of the area. However, for minerals, there are no such studies.
A R2 >0.25 correspond to a r >0.50. If used for genetic selection of beef population, the genetic correlations between the lab measured and the NIR predicted traits is often greater than the phenotypic correlation (i.e. the prediction correlation). So these NIRS prediction could be useful if used at abattoir level on large number of carcasses without sample up-taking for immediate results. Similarly they could be useful for monitoring quality of beef in the meat industries or as a screening technique for identifying meat cuts to be analyzed with more precise lab methods.
Re: The room temperature must be indicated with the value ± standard deviation. e.g. 23 ± 2.
AU: This value is corrected now. Line 115.

Reviewer 2 Report
The authors replied in all comments and the manuscript can be accepted in the present form
Author Response
Thank you very much for your effort and time. We appreciate your insightful comments and suggestions.